# Computer simulation approach to the identification of visfatin-derived angiogenic peptides

Ji Myung Choi[1,2☯], Srimai Vuppala[3☯], Min Jung Park[1,4], Jaeyoung Kim[3], Myeong-Eun Jegal[5], Yu-Seon Han[5], Yung-Jin Kim[5,6], Joonkyung Jang[3‡*], Min-Ho Jeong[2‡*], Bo Sun Joo[1,4‡*]

**1** Lab-to-Medi CRO Inc., Seoul, Republic of Korea, **2** Department of Microbiology, Dong-A University College of Medicine, Busan, Republic of Korea, **3** Department of Nanoenergy Engineering, Pusan National University, Busan, Republic of Korea, **4** The Korea Institute for Public Sperm Bank, Busan, Republic of Korea, **5** Korea Nanobiotechnology Center, Pusan National University, Busan, Republic of Korea, **6** Department of Molecular Biology, Pusan National University, Busan, Republic of Korea

☯ These authors contributed equally to this work.
‡ JJ, MHJ and BSJ also contributed equally to this work.
* jkjang@pusan.ac.kr (JJ); mhjeong@dau.ac.kr (MHJ); bosunoo@hanmail.net (BSJ)

**Data Availability Statement:** All relevant data are within the manuscript and its Supporting Information files.

## Abstract

Angiogenesis plays an essential role in various normal physiological processes, such as embryogenesis, tissue repair, and skin regeneration. Visfatin is a 52 kDa adipokine secreted by various tissues including adipocytes. It stimulates the expression of vascular endothelial growth factor (VEGF) and promotes angiogenesis. However, there are several issues in developing full-length visfatin as a therapeutic drug due to its high molecular weight. Therefore, the purpose of this study was to develop peptides, based on the active site of visfatin, with similar or superior angiogenic activity using computer simulation techniques. Initially, the active site domain (residues 181～390) of visfatin was first truncated into small peptides using the overlapping technique. Subsequently, the 114 truncated small peptides were then subjected to molecular docking analysis using two docking programs (HADDOCK and GalaxyPepDock) to generate small peptides with the highest affinity for visfatin. Furthermore, molecular dynamics simulations (MD) were conducted to investigate the stability of the protein-ligand complexes by computing root mean square deviation (RSMD) and root mean square fluctuation(RMSF) plots for the visfatin-peptide complexes. Finally, peptides with the highest affinity were examined for angiogenic activities, such as cell migration, invasion, and tubule formation in human umbilical vein endothelial cells (HUVECs). Through the docking analysis of the 114 truncated peptides, we screened nine peptides with a high affinity for visfatin. Of these, we discovered two peptides (peptide-1: LEYKLHDFGY and peptide-2: EYKLHDFGYRGV) with the highest affinity for visfatin. In an *in vitro* study, these two peptides showed superior angiogenic activity compared to visfatin itself and stimulated mRNA expressions of visfatin and VEGF-A. These results show that the peptides generated by the protein-peptide docking simulation have a more efficient angiogenic activity than the original visfatin.

**Funding:** This work was supported by the National Research Foundation of Korea (NRF) grant funded by the Korea government (MSIT) (No. 2020R1A2C2014089). The funders had no role in study design, data collection and analysis, decision to publish, or preparation of the manuscript.

**Competing interests:** The authors have declared that no competing interests exist.

## Introduction

The fundamental processes by which new blood vessels are formed in the body can be classified into vasculogenesis and angiogenesis. Vasculogenesis refers to the de novo formation of blood vessels from endothelial precursor cells in the body, and angiogenesis refers to the formation of new blood vessels from pre-existing vasculature. Angiogenesis is known to play an essential role in various normal physiological processes, such as embryogenesis, tissue repair, and organ regeneration. Angiogenesis also occurs in tumor cells to supplyblood for tumor growth and metastasis [1]. In addition, angiogenesis plays a very important role in skin regeneration, including wound healing and tissue recovery [2, 3], and in physiological events such as follicle development and embryo implantation [4–6].

Ovarian aging is a natural and physiological aging process characterized by a rapid decline in the oocyte quality along with a decrease in the number of follicles. It is a representative clinically unmet needfor infertility treatment. However, many recent studies have reported that ovarian aging can be reversed if an appropriate microenvironment is present in the ovary [7, 8]. One of the suggested strategies for overcoming ovarian aging is the activation of ovarian angiogenesis [9–12]. Our earlier studies have showed that the administration of visfatin during superovulation restores oocyte developmental competency and fertility potential in aged female mice [13, 14].

Visfatin was originally known as a pre-B cell colony-enhancing factor that promotes the growth of B-lymphocytes precursor cells [15]. However, it has recently been identified as an adipokine secreted from various cells such as macrophages, amniotic epithelial cells, human granulosa cells, and adipocytes [16–18]. In addition, since the first report of Kim *et al.* on the angiogenesis-promoting function of visfatin [19], many studies have shown that visfatin not only induces the production of the vascular endothelial growth factor (VEGF), a representative angiogenic factor, but also stimulates angiogenesis, including proliferation, migration, and metastasis of vascular endothelial cells [20–22].

However, high molecular weight of visfatin is a major limitation to its development as a therapeutic drug because full-length proteins have several inherent disadvantages, such as immunogenicity, lower stability, and loss of bioactivity [23]. In contrast, therapeutic peptides consisting of short-length amino acid sequence (3 to 20) have exhibited beneficial effects for the treatment of several health conditions [24, 25]. The peptides offer specific advantages such as good efficacy, good safety, low immunogenicity, high membrane permeability, and low cost compared to therapeutic proteins and antibodies [25–27]. For this reason, interest in the field of therapeutic peptides, including marine peptides, has increased in recent years. To date, more than 170 peptides are in active clinical development with many more in the preclinical stages [28–30]. In particular, recent improvements in peptide screening and computational biology have increased the demand for peptide drug discovery [25].

Most natural peptides act as agonists on their receptors, where the interaction between the peptide and the receptor is the first step to initiate an active response within the cell [31]. The protein-peptide docking can be simulated with computer-aided drug design (CADD) technologyand is very useful in deriving small agonist peptide sequences that can produce biological efficaciessimilar to that of natural ligands [32]. Therefore, the purpose of this study was to develop peptides,based on the active site of visfatin, with similar or superior angiogenic activity using computer simulation techniques.

## Materials and methods

### Construction of the peptide library

The following systematic approach was used to determine significant hotspots and active spots in the visfatin sequences. First, a three-dimensional structural analysis of visfatin showed that it was composed of two structural domains, and the active site was located in the second structural domain (from amino acid residues 181 to 390), which was defined as the active site domain (hereafter A-domain peptide) in this study [33, 34]. The A-domain peptide was then truncated into small peptides using the overlapping technique. Second, the three-dimensional structures of all the truncated peptide sequences were generated and predicted using the peptide tertiary structure prediction server with natural, non-natural,and modified residues (PEPstrMOD) [35, 36]. Finally, the designed 114 peptide sequences were subjected to molecular docking analysis to identify the hotspots in the globular A-domain peptide.

### Molecular docking simulations

A well-validated "structure-based molecular docking" computational technique that took into consideration the three-dimensional structure of visfatin was applied to design a small natural agonist peptide sequence. The three-dimensional structure of visfatin (PDB ID: 2G95) was obtained from the Research Collaboratory for Structural Bioinformatics Protein Data Bank (RCSB PDB)(http://www.rcsb.org), and the amino acids of the known active site of visfatin, namely, TYR18, PHE193, TYR195, ARG196, GLY197, ASP219, HIS247, ARG311, ARG313, GLY353, and ASP354, were identifed [33].The peptides with the highest affinity for visfatin were identified using two protein-peptide docking simulation programs, High Ambiguity Driven protein-protein DOCKing (HADDOCK) [37] and Galaxyprotein–peptide docking (GalaxyPepDock) [32].

### Molecular dynamics simulation of protein-ligand complexes

Molecular dynamics (MD) simulations were performed on the visfatin-peptide complexes to assess their structural stability. The Amber ff19SB force field [38] and optimal point charges (OPC) water model [39] were employed to model the protein and water molecules. Each system was neutralized in a 0.15 M NaCl solution by adding Na+ and Cl- ions. All the MD simulations were carried out in an NVT ensemble (fixed number of atoms, N, a fixed volume, V, and a fixed temperature, T) with a temperature fixed at 310 K. The Nose-Hoover thermostat method was used to keep the temperatureconstant. A leap-frog algorithm with a time step of 2 fs was used to propagate the MD trajectory. We run 50 ns of data production simulations without any restraints after 500 ps of equilibration simulation. The simulation progress was saved step by step every 100 ps. The simulation systems were constructed using Chemistry at Harvard Macromolecular Mechanics-Graphical User Interface (CHARMM-GUI) [40], and all the MD simulations were implemented using the Gromacs2021.5 software [41].

### *In silico* toxicity, angiogenic, antihypertensive profiles, and half-life predictions of peptides

The toxicity profiles (hemotoxicity, cytotoxicity, and immunotoxicity) of the designed peptides were predicted using the *in silico* tools ToxinPred [42] and HemoPI [43]. Machine learning using support vector machine and quantitative matrix (QM) models was used to predict the toxicity of the peptides. The half-life of a peptide is related to its stability and toxicity. The PlifePred [44] and HLP [45] web servers were used to predict the half-lives of peptides in blood and intestine-like environments, respectively. The AHTPin [46] and

AntiAngiopred [47] web servers were used to predict the antiangiogenic and hypertensive properties of the designed peptides.

## Preparation of visfatin and visfatin-derived peptides for cytotoxicity and angiogenesis assessment *in vitro*

The two synthetic peptides (peptide-1 and peptide-2) derived from the docking simulation having the highest affinity were obtained from GL Biochem Ltd. (Shanghai, China) and used for cytotoxicity and angiogenesis assessment *in vitro*. The synthetic peptides were supplied as dry powders with a purity of 95% or more guaranteed by mass spectrometry and high-performance liquid chromatography (HPLC), reconstituted in dimethyl sulfoxide (DMSO) to a concentration of 1 mg/mL, and stored at -20˚C until use. Commercial recombinant mouse visfatin were purchased from Adipogen Life Science, Inc. (San Diego, USA). For preparing the stock solution, visfatin was dissolved in DMSO to a concentration of 1 mg/mL, and recombinant human basic fibroblast growth factor (b-FGF) was dissolved in phosphate-buffered saline (PBS) to a concentration of50 μg/ml. These stock solutions were stored at -20˚C until use.

Prior to cytotoxicity and angiogenesis assay experiments, b-FGF, visfain, and visfatin-derived peptides were diluted with endothelial cell growth medium (EGM) supplemented with an EGM-Supplement Mix® (PromoCell) to achieve the working concentration. b-FGF and visfain were used as a positive control, and the EGM was used as a control. The EGM-Supplement Mix® consisted of 2% fetal bovine serum (FBS), 0.004 ml/ml endothelial cell growth Supplement, 0.1 ng/ml recombinant human epidermal growth factor (rhEGF), 1 ng/ml recombinant human b-FGF, 90 μg/ml heparin, and 1 μg/ml hydrocortisone.

## Culture and maintenance of human vascular endothelial cells (HUVECs)

Human vascular endothelial cells (HUVECs) (PromoCell, Heidelberg, Germany) were cultured inendothelial cell growth medium 2 (EGM-2, PromoCell) supplemented with EGM-2 Supplement Mix®(PromoCell) and 2% fetal bovine serum (FBS) for 24 hours at 37˚C and 5% $CO_2$. The EGM-2 Supplement Mix® consisted of 5 ng/ml rhEGF, 10 ng/ml recombinant human b-FGF, 20 ng/ml recombinant human insulin-like growth factor (Long $R_3$rhIGF,), 0.5vng/ml recombinant human vascular endothelial growth factor 165 (VEGF), 1 μg/ml ascorbic acid, 22.5 μg/ml heparin, 0.2 μg/ml hydrocortisone. Cells between passages 2 and 7 were used in all experiments.

## MTT cell cytotoxicity assay

HUVECs were grown in EGM-2 supplemented with an EGM-2 Supplement Mix® (PromoCell) containing 2% FBS at a density of $1 \times 10^4$ cells in 96-well plates (SPL Inc., Gyeonggido, Korea). After 24 hours, the cells were treated with 1000ng/ml visfatin (Adipogen Life Science) and various concentrations of visfatin-derived peptides (0.1, 0.5, 1.0, and 2.0 μM), and then incubated overnights in an atmosphere of 5% $CO_2$. The 3-(4,5-Dimethylthiazol-2-yl)-2,5-diphenyltetrazolium bromide (MTT) assay reagents (Sigma-Aldrich, St. Louis, MO, USA) were as added directly to each well to a final concentration of 0.5 mg/ml. After 4 hours, the medium was removed, the formazan crystals formed in the cells were dissolved in DMSO, and the absorbance of the formazan solution was measured using a Synergy HTX Multi-Mode Reader (BIO-TEK, Vermont, USA) with a 540 nm filter. Each sample was assayed in triplicate.

## *In vitro* trans-well invasion assay

The invasion capacity of the cells was determined using a 24-well trans-well system. The upper side of the trans-well membrane was coated with 1 mg/mL Matrigel at 10 μL/well at room temperature for 1 hour. The EGM-2 medium (0.5 ml)was added to the lower part of trans-well chamber. HUVECs ($2 \times 10^4$ cells) and EGM medium (0.1 ml) were placed on the upper compartment of the trans-well. And then b-FGF (25ng/ml), visfatin (1000 ng/ml), and visfatin-derived peptides (0.5μM and 2.0μM) were added to the upper compartment of the trans-well. Cells were incubated at 37˚C and 5% $CO_2$ for 24 hours, fixed with methanol, and stained with hematoxylin and eosin. The cells on the upper surface of the membrane were removed by wiping with a cotton swab. Cell invasion was determined by counting the number of whole cells in a single filter using optical microscopy at 40× magnification. Each sample was assayed in duplicate, and independent experiments were repeated three times.

## *In vitro* wound healing migration assay

HUVECs were seeded in 24-well plates (SPL) at a density of $2 \times 10^5$ cells and incubated in EGM medium supplemented with EGM-Supplement Mix® containing 1% FBS overnight. Cells were scratched using a P200 pipette tip to induce a wound. Subsequently, then b-FGF (25ng/ml), visfatin (1000 ng/ml), and visfatin-derived peptides (0.5μM and 2.0μM) were added and the culture was maintained for 18 hours to allow the cells to migrate. The migration patterns were observed using a phase-contrast microscope and images were captured. The wound diameters were photographed at 16–24 hours. Wound closure was determined using optical microscopy at 40× magnification. Migration was quantified by counting the number of cells that had moved beyond the reference line.

## *In vitro* tube formation assays

HUVECs ($2 \times 10^4$ cells/well) were seeded on a layer of polymerized Matrigel and treated with or without visfatin-derived peptides. The Matrigel culture was incubated at 37˚C in the EGM supplemented with EGM-Supplement Mix® containing 1% FBS. After 4 hours, changes in the cell morphology were observed using a phase-contrast microscope and photographed at 40× magnification. Each sample was assayed in duplicate, and independent experiments were repeated three times. Angiogenesis was analyzed with ImageJ software using an angiogenesis analyzer (Bethesda, MD, USA).

## Real-time quantitative PCR (RT-qPCR)

After treating HUVECs with visfatin-derived peptides for 24 hours, the total RNA was isolated from the cultured HUVECs using the Tri-RNA reagent (Favorgen Biotech Corp., Kaohsiung, Taiwan) according to the manufacturer's instructions. cDNA was synthesized from 1 μg of total RNA using the iScript™ cDNA Synthesis Kit (Bio-Rad Laboratories, Inc., Hercules, CA, USA). The mRNA expression levels of VEGF and visfatin were carried out using Applied Biosystems 7500 software (Applied Biosystems Inc., Waltham, MA, USA). Real-time PCR was performed using SYBR Green Q-PCR Master Mix (Samjung Bio Science, Daejeon, South Korea). The reaction conditions were as follows: initial denaturation at 95˚C for 5 min, amplification at 95˚C for 15 sec, and at 60˚C for 1 min. All PCR amplifications were performed for 40 cycles. The specific primers used were as follows: 5'-AAGGGTCATCATCTCTGCCC-3' (forward) and 5'-GTG ATGGCATGGACTGTGGT-3' (reverse) for glyceraldehyde-3-phosphate dehydrogenase (GAPDH); 5'-GGAGG GTGACGGGGTGAAGG-3' (forward) and

5′-GTCGGTGGCCAGGAGGATGTT-3′ (reverse) for visfatin; and 5′-CTACCTCCACCATG CCAAGT-3′ (forward) and 5′-AGCTGCGCTGATAGACATCC-3′ (reverse) for VEGF-A. The relative changes in the gene expression were determined using the $2^{-\Delta\Delta Ct}$ method and the results were normalized to the expression of GAPDH.

### Statistical analysis

All data are presented as mean ± standard deviation (SD). The Prism 9 program (GraphPad Inc., USA) was used for statistical analysis. The experimental results were statistically processed using the t-test and two-way ANOVA. Statistical significance was set at P value < 0.05.

## Results

### Construction of peptide library using the overlapping technique

To find a hotspot or active spot in the visfatin sequence, the A-domain (residues 181–390) of visfatin was truncated into small peptidesutilizing the overlapping technique. As a result, 114 truncated small peptides were generated and categorized into three datasets based on sequence length 6 (dataset-1, 38 peptides), 7 (dataset-2, 42 peptides), and 9 (dataset-3, 34 peptides) with overlapping amino acids 3, 2, and 3, respectively (Fig 1).

### Molecular docking simulationsof small peptides

Two protein-peptide docking simulation programs, *viz.*HADDOCK and GalaxyPepDock were used to perform the molecular docking simulations of the 114 truncated peptides. The HADDOCK score is a measure of how well the peptide interacts with the active site and represents its binding capability. On the other hand, GalaxyPepDock performs similarity-based docking by finding templates from the database of experimentally determined structures and building models. The GalaxyPepDock score is thus used to evaluate similarity of interactions in newly designed protein-peptide complexes to those observed in established protein-peptide complexes found in a database, thus providing insights into their compatibility and potential binding characteristics. From the docking analysis of the 114 peptides, 27 peptide sequences showed high scores with a sequence length of 9 (dataset-3). Therefore, we considered a peptide sequence length of 9 as the standard length for further screening. From the 27 peptide sequences, we reconstructed and identified six hotspots "LEYKLHDFGYRGVSSQ, GIALIKKY YGTKDPV, IYNACEKIWGEDLRH, HSTITAWGKDHEKDAF, KFPVSENSKGYKLLPPY, and GMKQKKWSIENVSFG" in the A-domain peptide of visfatin (Fig 1) and they were considered for further analysis to identify the best agonistic peptides.

Seventy-six peptide sequences with different lengths from the six hotspots were designed and subjected to docking simulation. To understand the interactions of our designed peptides, we performed the molecular docking simulations of nicotinamide mononucleotide (NMN) which is a natural ligand (agonist) of visfatin(Fig 2A). The active site residues, GLY384, with an H-bond length (H-BL) of 2.52 Å, ARG196 (H-BLs 2.63 Å and 2.56 Å), and ARG311 (H-BL 2.63 Å) formed H-bond interactions with NMN, signifying their importance in the catalytic activity. The catalytic site interactions of nine peptides among those screened (Table 1) were similar to those of natural NMN ligand. This result indicates that the nine peptides may act as potential agonists of visfatin. Of the nine peptides, peptide-1 (LEYKLHDFGY, 10AA) and peptide-2 (EYKLHDFGYRGV, 12AA) had the highest scores in the two docking programs. Fig 2B and 2C shows the molecular interactions of the two peptides within the active site of visfatin. Further, the stability of the protein-peptide complexes was confirmed from the MD simulations under solvent conditions.

MNAAAEAEFNILLATDSYKVTHYKQYPPNTSKVYSYFECREKKTENSKVRKVKYEETVFYGLQYILNKYL
KGKVVTKEKIQEAKEVYREHFQDDVFNERGWNYILEKYDGHLPIEVKAVPEGSVIPRGNVLFTVENTDPEC
YWLTNWIETILVQSWYPITVATNSREQKKILAKYLLETSGNLDGLEYKLHDFGYRGVSSQETAGIGASAHLV
NFKGTDTVAGIALIKKYYGTKDPVPGYSVPAAEHSTITAWGKDHEKDAFEHIVTQFSSVPVSVVSDSYDIYN
ACEKIWGEDLRHLIVSRSTEAPLIIRPDSGNPLDTVLKVLDILGKKFPVSENSKGYKLLPPYLRVIQGDGVDI
NTLQEIVEGMKQKKWSIENVSFGSGGALLQKLTRDLLNCSFKCSYVVTNGLGVNVFKDPVADPNKRSKKG
RLSLHRTPAGTFVTLEEGKGDLEEYGHDLLHTVFKNGKVTKSYSFDEVRKNAQLNMEQDVAPH

**Fig 1. Fasta sequence of active site domain (A-domain) of visfatin (491AA).** Shown in blue and red is the sequence belonging to the A-domain of visfatin, respectively. Six hotspots are shown in red.

## Molecular dynamic simulations of the visfatin-derived peptides

The molecular dynamic (MD) simulations were performed for peptide-1 and peptide-2 using Gromacs2021.5 software. The highest-scoring conformations of the protein-ligand complex, obtained through molecular docking, were chosen for the MD simulations to evaluate the system's stability under solvent conditions. The final trajectory of equilibration was used as the initial structure for the data production simulation. We calculated the "root mean square deviation" (RMSD), and "root mean square fluctuations" (RMSF) of peptide-1 and peptide-2. The RMSD plot of the two peptides was shown in Fig 3. Starting from an initial value, the RMSD of peptide-1 showed a gradual increase during the course of the simulation. However, at 9.4 ns, the rate of increase accelerated significantly, ultimately reaching 5.18 Å by 14.7 ns. After that, the peptide-1 was stabilized in the active site of the visfatin during the simulation time of 50 ns (Fig 3A). The RMSD plot of peptide-2 showed the fluctuations up to 2ns after which the peptide gradually attained a stable position and large fluctuations were not observed during the simulation time (Fig 3B).

The RMSF plot was generated for two protein-peptides complexes to estimate the internal fluctuations in the protein and ligands. In the case of the peptide-1 complex, noticeable

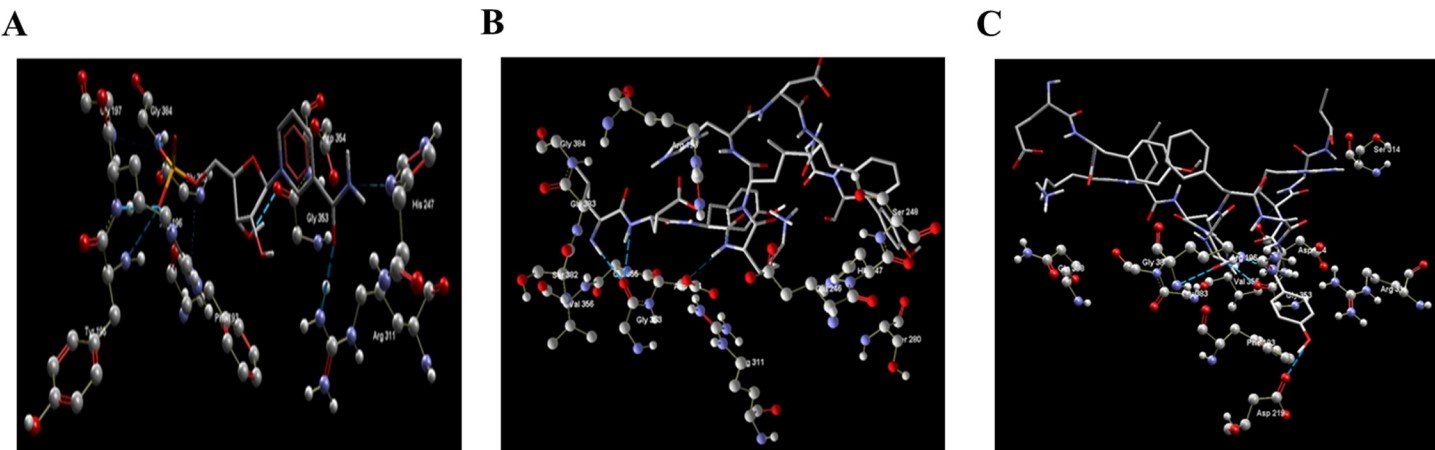

**Fig 2.** Diagram of the binding pose of (A). NMN, (B). Peptide-1 (LEYKLHDFGY), and (C). Peptide-2 (EYKLHDFGYRGV). The NMN, peptide-1 and peptide-2 are drawn as sticks, and the visfatin active site residues are represented by balls and sticks. The blue-colored broken lines represent H-bond interactions.

**Table 1. Molecular docking results of the peptide sequences similar to NMN against visfatin.**

| | Amino acid sequence | HADDOCK Score | GalaxyPepDock score |
|---|---|---|---|
| Peptide-1 | LEYKLHDFGY(10AA) | -106.0 +/- 2.0 | 16.0 |
| Peptide-2 | EYKLHDFGYRGV(12AA) | -105.9 +/- 3.5 | 17.0 |
| Peptide-3 | ALIKKYYGTKDPV (13AA) | -111.2 +/- 2.9 | 10.0 |
| Peptide-4 | STITAWGKDHEKDAF (15AA) | -124.3 +/- 6.5 | 1.0 |
| Peptide-5 | ACEKIWGEDLRH (12AA) | -85.5 +/- 5.5 | 0.0 |
| Peptide-6 | CEKIWGEDLRH (11AA) | -93.6 +/- 4.2 | 0.0 |
| Peptide-7 | EKIWGEDLRH (10AA) | -71.6 +/- 2.0 | 0.0 |
| Peptide-8 | GMKQKKWSIENVSF (14AA) | -97.8+/- 4.3 | 15.0 |
| Peptide-9 | ENSKGYKLLPPY (12AA) | -82.7 +/- 2.9 | 18.0 |

fluctuations were observed at residue positions 1 (0.5 nm) and 4 (0.42 nm) of the peptide, whereas, for peptide-2, no significant fluctuations were observed, with an RMSF value of less than 0.25 nm. According to the data presented in Fig 4A, a prominent fluctuation was observed in the visfatin chain at residue position 255, with a value of 0.55 nm. However, no significant fluctuations were observed for the remaining segments of the chain,, and the RMSF value remained below 0.25 nm. In the visfatin- peptide-2 complex (Fig 4B), no large fluctuations were observed and the RMSF value was less than 0.25 nm which meantthat both the visfatin and peptide were stable throughout the simulation time of 50 ns.

### *In silico* toxicity, angiogenic, antihypertensive, and half-life predictions of peptides

The *in silico* toxicity analysis showed that two peptides (peptide-1 and peptide-2) were non-toxic, with support vector machine (SVM) and quantitative matrix (QM) scores of lessthan zero (S1 and S2 Tables). The hemolytic prediction of these two peptides with an SVM score of zero represents non-hemolytic properties (S1 Table). The half-life of the designed peptides was less than 14 minutes, offering normal stability and negligible toxicity (S3 Table). These two peptides were non-anti-angiogenic and non-antihypertensive, with an SVM score of less than zero (S4 Table).

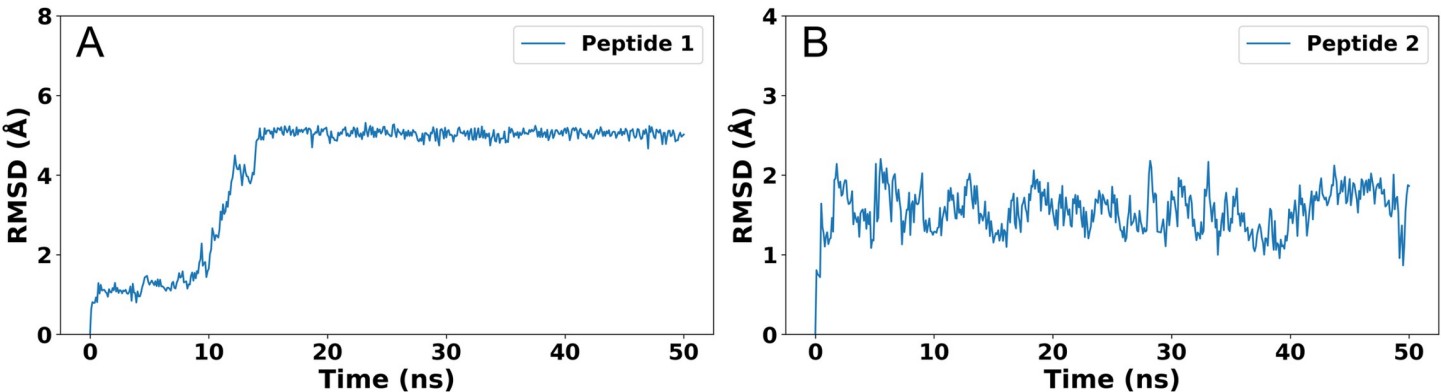

**Fig 3. The RMSD plots of peptide-1 and -2 with respect to the initial complex during 50 ns of simulation time.** (A) Peptide-1, (B) Peptide-2. The blue color trajectories represent the RMSD evolution of peptides.

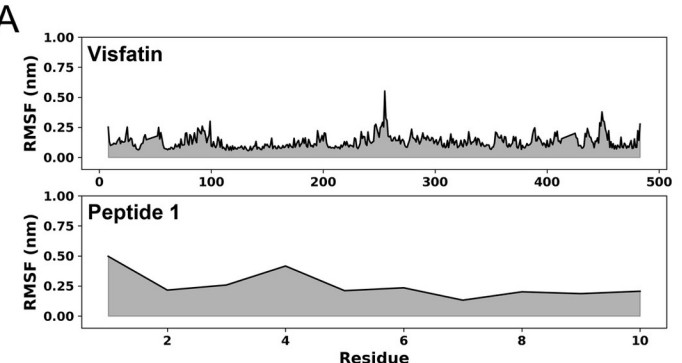
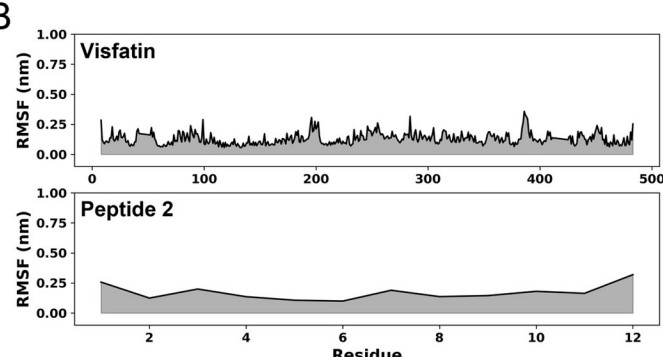

**Fig 4. The RMSF plots of visfatin-peptide complexes.** (A) Visfatin and peptide-1, (B) Visfatin and peptide-2. The black color trajectories represent the fluctuations in the visfatin and peptides.

### Effect of the two peptides on cytotoxicity

To investigate the cytotoxic effect of peptide-1 and -2, each peptide was measured in an MTT assay at concentrations of 0.1, 0.5, 1.0, and 2.0 μM. Treatment with the two peptides for 24 hours did not affect the viability of HUVECs (Fig 5). This indicated that peptide-1 and peptide-2 had no cytotoxic effect on HUVECs.

### Effects of the two peptides on angiogenesis

The role of peptide-1 and -2 in promoting angiogenesis *in vitro* by invasion, migration, and tube formation in the HUVECs was examined at concentrations of 0.5 and 2.0 μM. As shown in Fig 4, cell invasion was significantly promoted more than two-fold by both peptides compared to the control group. Specifically, 0.5 μM peptide-1 and 2.0 μM peptide-2 increased cell invasion more than two-fold compared to the control and visfatin-treated groups. The increase in cell invasion by the peptides was higher than that by b-FGF in the positive control group (Fig 6).

Treatment with the two peptides resulted in a two-fold increase in cell migration at concentration of 0.5 μM for peptide-1 and 2.0 μM for peptide-2 compared to the control group (Fig 7).

Angiogenic tube formation on Matrigel is an indication of the formation of blood vessels, such as capillaries, and is measured by two factors, mesh and master junction formation. The number of meshes was not significantly different when b-FGF and visfatin were used as positive controls compared to the control group, but was significantly increased by treatment with 0.5 μM concentration of peptide-1 and -2 (P<0.01, P<0.001, respectively). In contrast, the number of master junctions was significantly increased in both peptide-1 and -2 except at the 2.0 μM concentration of peptide-2. This increase was greater than that of b-FGF and visfatin (Fig 8).

### Effects of the two peptides on the mRNA expression of VEGF and visfatin

To confirm whether the two peptides stimulate the expression of visfatin and VEGF, the mRNA expressions of visfatin and VEGF were evaluated in the HUVECs treated with the two peptides. The expressions of both visfatin and VEGF-A were significantly increased at 0.5 μM of peptide-1 and at both 0.5 and 2.0 μM of peptide-2 compared to the control group (P<0.05). These expressions were comparable to those of visfatin and b-FGF treatment. Specifically, the VEGF-A expression was remarkably increased at 2.0 μM of peptide-2. On the other hand, 2.0 μM of peptide-1 did not increase the expression of visfatin and VEGF-A (Fig 9).

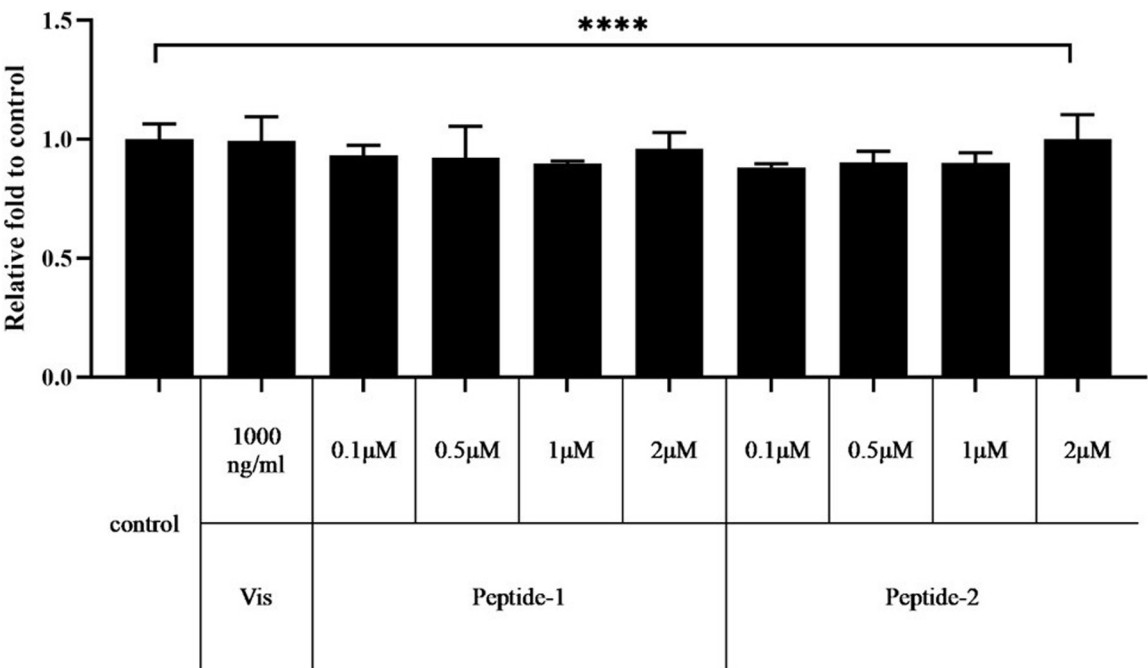

**Fig 5. Cytotoxic effects of peptide-1 and -2 on HUVECs cells.** Cell viability was estimated 24 hours after peptide-1 and -2 treatment using an MTT assay. Data are presented as mean±SD of three independent experiments. Vis: visfatin.

## Discussion

This study showed that the two peptides derived from visfatin by protein-peptide docking simulations have more efficient angiogenic activity than visfatin itself. In addition, this study revealed that the two peptides stimulated the expression of VEGF-A and visfatin. This is the

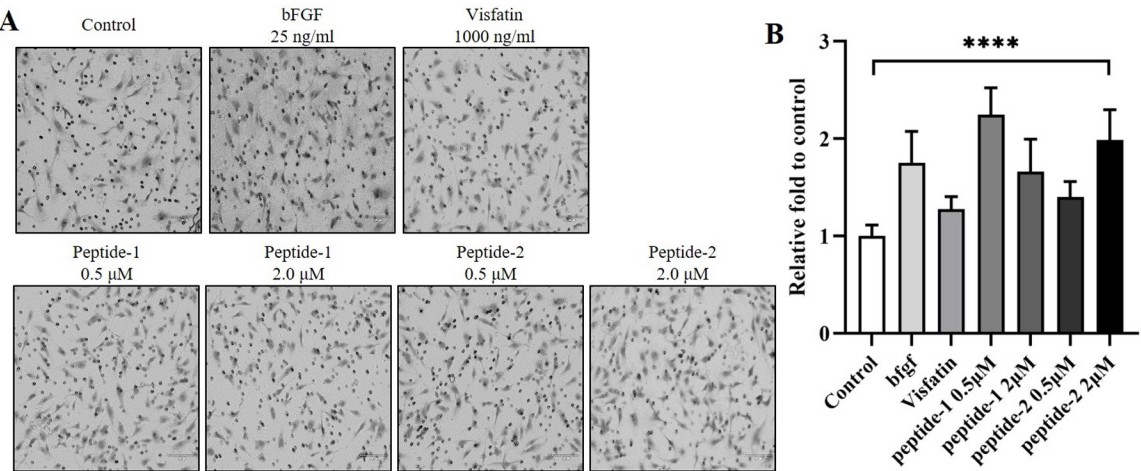

**Fig 6. Cytotoxic effects of peptide-1 and −2 on cell invasion.** (A) Representative image and (B) graph of the invaded cells after 24 hours. HUVECs ($2 \times 10^4$) were seeded in 100 μL of serum-free media with visfatin peptides added to the upper compartment of the transwell and the full medium was added to the lower compartment. After 24 hours, cell invasion was determined by counting the number of whole cell in a single filter using optical microscopy (40×). Data are presented as mean±SD of three independent experiments. ****P<0.0001 (vs control).

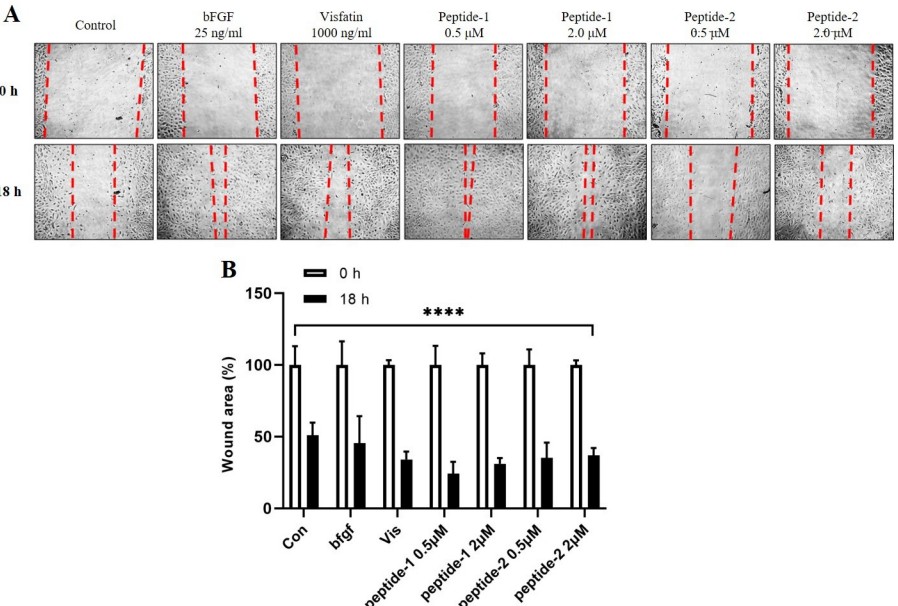

**Fig 7. Effects of peptide-1 and -2 on cell migration.** (A) Representative image and (B) graph of the migrated cells at 18h. HUVECs ($2\times10^5$/well) were seeded on 24-well plates and incubated overnight. Then, the cells were scratched using a P200 pipette tip and further incubated in media with or without peptide-1 and -2. Cells were allowed to migrate for 18h. Migration patterns were observed using a phase-contrast microscope (×40). Data are presented as mean±SD of three independent experiments. ****$P<0.0001$ (vs control).

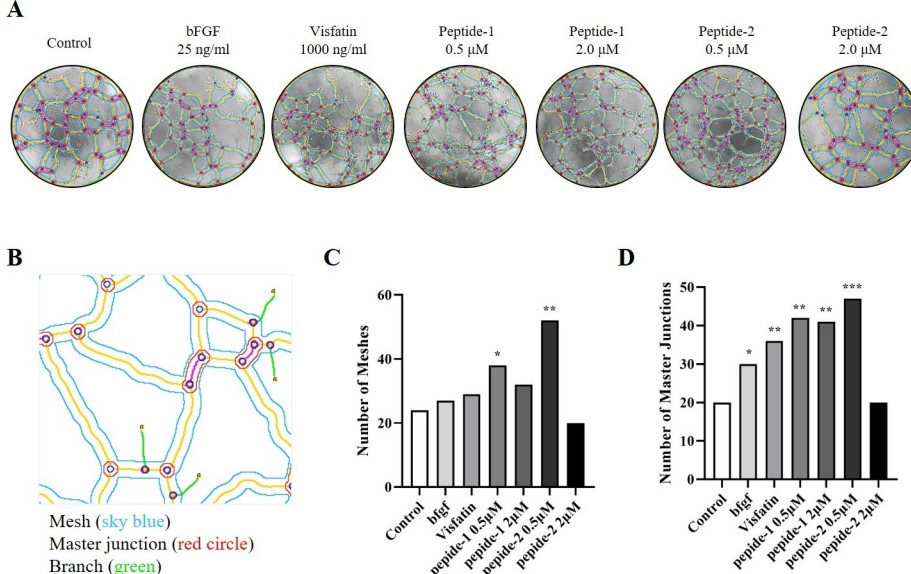

**Fig 8. Effect of peptide-1 and -2 on tube formation on Matrigel assays.** (A) Representative tube formation images, (B) schematic image of angiogenesis, (C) number of branches, and (D) number of total branches. HUVECs ($2\times10^4$/well) were seed on a layer of previously polymerized Matrigel and treated with or without peptide-1 and -2. The Matrigel culture was incubated at 37°C. After 4 hours, changes in cell morphology were captured using a phase-contrast microscope (×40). Each sample was assayed in duplicate, and independent experiments were repeated three times. *$P<0.05$, **$P<0.01$, and ***$P<0.001$ (vs control).

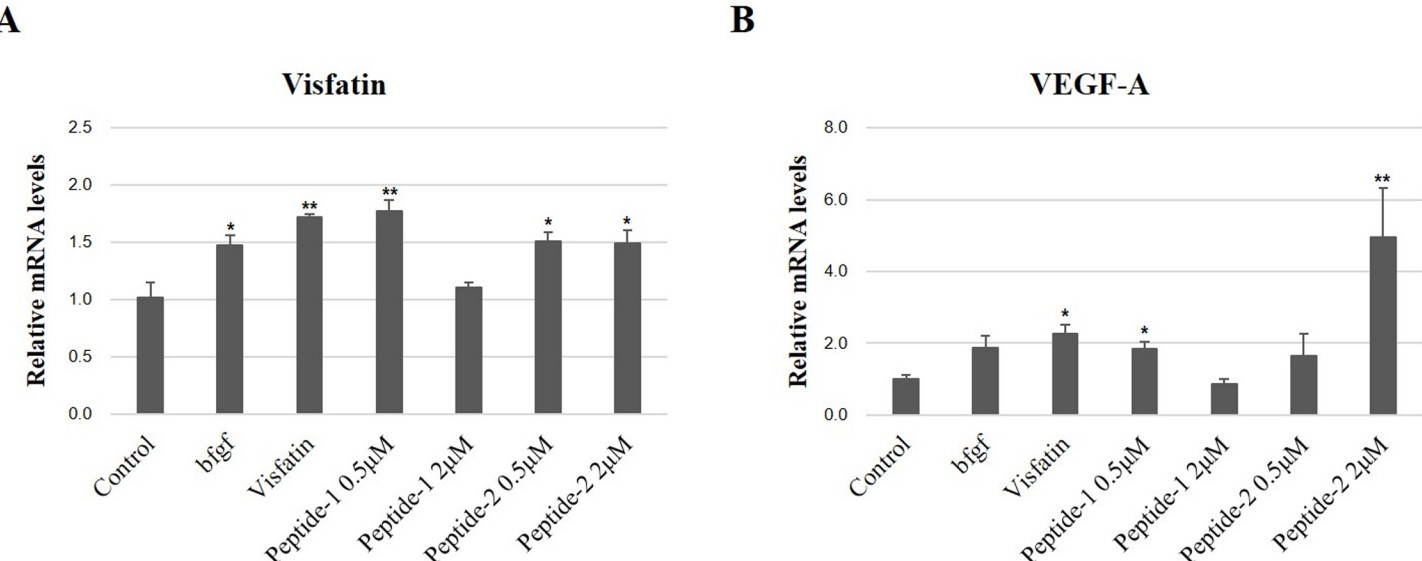

**Fig 9. Effect of peptide-1 and -2 on expression of visfatin and VEGF-A.** mRNA expressions of visfain and VEGF-A was examined in two peptides-treated cells by real-time quantitative PCR (RT-qPCR) analysis Each PCR was performed in triplicate for each sample. Relative gene expression levels were calculated versus GAPDH. Data are presented mean±SD. *P<0.05 and **P<0.01 (vs control). b-FGF: Basic fibroblast growth factor.

first report on the development of visfatin-derived peptides with better efficacy than the original full-length visfatin using computational techniques.

It is well known that VEGF and periostin to stimulate angiogenesis. Therefore, some studies have reported the identification of VEGF- or periostin-derived angiogenic peptides containing about 10~15 amino acids as agonists to overcome the disadvantages of using full-length proteins [48–50]. However, to our knowledge, no studies on visfatin-derived angiogenic peptides carried out to date. In addition, the amino acid sequences of peptide-1 and -2 identified in the present study have not been reported earlier and hence this study may be considered a novel study.

The present study took into consideration two concepts to design visfatin agonist based on the active site. First, the designed peptide should act as a natural ligand of visfatin and produce similar biological effects when it binds to the receptor. Second, the designed peptide should be a visfatin mimetic. It was not possible to identify the protein-ligand binding structure using experimental methods because this would require considerable time and resource without the assistance of computer technology. However, as many computer-based docking simulation programs have been developed after the geometrical study of macromolecular-ligand interactions by Kuntz et al. [51], the study of protein-ligand interactions has now become very easy.

Predicting the binding affinity between a ligand and a specific target protein is important in drug design. Molecular docking simulation is an excellent technique for identifying the optimal ligand-binding conformation for a specific target [52, 53]. For this purpose, two docking simulation programs, HADDOCK [37] and GalaxyPepDock [32] were used in this study. Through simulations, these two programs provide a score for the binding affinity between the ligand and the target protein or receptor. Among the 114 peptides obtained by the overlapping technique, peptides with an amino acid sequence suitable for the visfatin active site were evaluated using the HADDOCK docking simulation. The interaction between each peptide and visfatin was evaluated based on the similarity of the interaction between the ligand and the target protein stored in the GalaxyPepDock docking program.

In this study, the overlapping technique was used prior to docking simulation to design the peptide. This strategy reduces the efforts required to synthesize a peptide and to analyze the interaction with the receptor. As a result, 114 peptides were generated from the truncated A-domain of viafatin and used in the docking simulation.

The analysis of interactions between natural ligands and target proteins or receptors is very importantin peptide design [31, 54]. NMN is a natural ligand of visfatin that acts as an agonist [55]. This study simulated the docking of the natural NMN ligand (Fig 2) against visfatin to understand the catalytic site interactions. The results confirmed that the catalytic site interactions of nine peptides among those screened (Table 1) were similar to those of the natural NMN ligand.

Since the biological unit of visfatin is dimer, it is also important to consider the dimer conformation for peptide design. Hence, the present study performed molecular docking simulation to design peptides by considering visfatin as a dimer in the GalaxyPepDock software and as a monomer in the HADDOCK software, and calculated their corresponding scores. As shown in the RMSD (Fig 3) and RMSF (Fig 4) plots, peptide-1 had more fluctuations than peptide 2. During the initial stages of the dynamics simulation, the N-terminus (residue position 1–4) of peptide 1 was weakly bound to visfatin, but the binding was disrupted at 9.4 ns. Subsequently, the structure of the N-terminus underwent changes and eventually reached a stable conformation after 14.7 ns. This resulted in fluctuations in the RMSF and RMSD plots. On the other hand, the C-terminus (residue position 6–10), which primarily interacts with the active site of visfatin, did not show significant structural changes during the simulation. Therefore, it appears that the structure of the visfatin-peptide1 complex is stable. In the case of peptide-2, the peptide chain was located inside the active site of visfatin, and fluctuated less during the simulation.

Experimental determination of the toxicity of a large number of peptides is laborious, costly, and time-consuming. *In silico* toxicity analysis may provide an alternative to tedious experimental methods for predicting the toxicity of peptides. Based on our *in silico* toxicity predictions, out of the nine designed peptides, two peptides (peptide-1: LEYKLHDFGY and peptide-2; EYKLHDFGYRGV) were found to be non-toxic. In general, the half-life of an ideal peptide should be less than a few hours for it to stabilize and activate the immune system [56]. In this respect, these two peptides have normal stability and less toxicity with half-lives of less than 14 minutes. In addition, these two peptides were non-anti-angiogenic and non-antihypertensive. This study also evaluated the mRNA expression levels of visfatin to confirm the agonist activity of the designed peptides from *in silico* analysis. In addition, we evaluated the mRNA expression levels of VEGF and the angiogenesis efficacy of these peptides using *in vitro* analysis.

In evaluating the angiogenesis efficacy of the two peptides (peptide-1 and -2), the determination of the treatment concentration also required careful consideration. Many previous studies had used concentrations in the range from 0.1 to 25 μM in the evaluation of angiogenesis activity of designed peptides [57, 58]. The present study carried out cytotoxicity assay *in vitro* at concentration range from 0.5 to 2.0 μM, and no cytotoxicity was observed at these concentrations. Based on these results, the lowest concentration of the peptides, at which no cytotoxicity was observed, was used to evaluate angiogenesis efficacy.

One of the limitations of the present study is that it is uncertain whether the synthesized peptide had a configuration similar to visfatin as the same set of amino acids also had different configurations. Hence, we additionally evaluated the conformation and configuration of the synthetic peptides using circular dichroism (CD) spectroscopy. As shown in S1 Fig, peptide-1 was found to display two peaks (around 200 nm and 225 nm), and peptide-2 displayed a slight peak around 225 nm. These CD results did not show a typical spectrum of a-helix, b-sheet,

and disordered form and therefore, we could not evaluate the configuration of the peptides. Nevertheless, the synthesized peptides may contain an optical isomer with the same configuration as that found in natural visfatin, and should have similar biological activities. In the future study, it would be important to consider the peptide configuration when designing peptides for therapeutic applications.

## Conclusions

In conclusion, this study identified and developed two peptides as visfatin agonists, which stimulated the mRNA expression levels of visfatin and VEGF using *in silico* and *in vitro* analysis. Most importantly, these two peptides exhibited superior angiogenesis activity compared to visfatin itself. The peptides introduced in this study were developed for the first time and hold promise for facilitating the creation of therapeutic agents aimed at promoting skin regeneration and enhancing ovarian function.

## Supporting information

**S1 Table. Toxicity and hemolytic activity predictions of designed peptides in different models using the SVM method.**
(DOCX)

**S2 Table. Toxicity predictions of designed peptides in different models using the QM method.**
(DOCX)

**S3 Table. Half-life of Peptides (HLP) in intestine-like environment and blood (Seconds).**
(DOCX)

**S4 Table. Angiogenic and hypertensive activities of designed peptides.**
(DOCX)

**S1 Fig.** Circular dichroism (CD) spectroscopyof peptide-1 (A) and peptide-2 (B).
(TIF)

## Author Contributions

**Conceptualization:** Ji Myung Choi, Srimai Vuppala, Min Jung Park, Bo Sun Joo.

**Data curation:** Jaeyoung Kim, Myeong-Eun Jegal, Yung-Jin Kim.

**Investigation:** Min Jung Park.

**Methodology:** Srimai Vuppala, Min Jung Park, Jaeyoung Kim, Myeong-Eun Jegal, Yu-Seon Han, Yung-Jin Kim.

**Software:** Joonkyung Jang.

**Supervision:** Joonkyung Jang, Min-Ho Jeong, Bo Sun Joo.

**Writing – original draft:** Ji Myung Choi, Srimai Vuppala, Min Jung Park.

**Writing – review & editing:** Joonkyung Jang, Min-Ho Jeong, Bo Sun Joo.

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
