## [Decision Letter · Decision Letter 0]

5 May 2023

PONE-D-23-06800Computer simulation approach to the identification of peptides with angiogenic activity based on the visfatin active sitePLOS ONE

Dear Dr. Joo,

Thank you for submitting your manuscript to PLOS ONE. After careful consideration, we feel that it has merit but does not fully meet PLOS ONE’s publication criteria as it currently stands. Therefore, we invite you to submit a revised version of the manuscript that addresses the points raised during the review process.

We look forward to receiving your revised manuscript.

Kind regards,

Belgin Sever, Ph.D.

Academic Editor

PLOS ONE

“This work was supported by the National Research Foundation of Korea(NRF) grant funded by the Korea government (MSIT) (No. 2020R1A2C2014089)”

“NO”

Reviewers' comments:

Reviewer's Responses to Questions

**Comments to the Author**

1. Is the manuscript technically sound, and do the data support the conclusions?

Reviewer #1: No

Reviewer #2: Yes

2. Has the statistical analysis been performed appropriately and rigorously? 

Reviewer #1: No

Reviewer #2: Yes

3. Have the authors made all data underlying the findings in their manuscript fully available?

Reviewer #1: Yes

Reviewer #2: Yes

4. Is the manuscript presented in an intelligible fashion and written in standard English?

Reviewer #1: No

Reviewer #2: Yes

5. Review Comments to the Author

Reviewer #1: After careful examination of this manuscript entitled " Computer simulation approach to the identification of peptides with angiogenic

activity based on the visfatin active site" written by Choi et.al, I observed that this manuscript unsuitable to publish in this journal.

Comments:

1. There was no novelty in this study.

2. MD simulation should be required for docking validation

3. Result and dicsussion section was not clear insight.

4. Results were not consistent with previous studies.

5. The biological unit of visfatin is dimer and why author did not consider dimer conformation for peptide design?

Reviewer #2: In the manuscript (PONE-D-23-06800), authors researched the visfatin-derived angiogenic peptides by computer simulation approach. Overall, the topic is of scientific interest and would be of interest in the field of bioactive peptides. I think the manuscript can be accepted and published in PLOS ONE after a major revision. Suggestions are provided below:

(1) Title: Should be “Computer simulation approach to the identification of visfatin-derived angiogenic peptides” rather “Computer simulation approach to the identification of peptides with angiogenic activity based on the visfatin active site”.

(2) Keywords: Small pPeptide is not a standard term. I think that “oligopeptide” might be better than “small peptides”.

(3) Section Abstract

--Line 34: Should be “Firstly” rather “First”.

--Line 35: Should be “was” rather “was first”.

--Line 36: Should be “Secondly” rather “Second”.

(4) Section Introduction: The introduction provides an adequate background on the topic.

--Line 79-81: Authors are advised to add references.

[1] Sridhar K, Inbaraj BS, Chen BH. Recent developments on production, purification and biological activity of marine peptides. Food Res Int. 2021, 147, 110468.

[2] Suo SK, Zheng SL, Chi CF, Luo HY, Wang B. Novel angiotensin-converting enzyme inhibitory peptides from tuna byproducts-milts: Preparation, characterization, molecular docking study, and antioxidant function on H2O2-damaged human umbilical vein endothelial cells. Front. Nutr. 2022, 9, 957778.

[3] Sheng Y, Wang WY, Wu MF, Wang YM, Zhu WY, Chi CF, Wang B. Eighteen Novel Bioactive Peptides from Monkfish (Lophius litulon) Swim Bladders: Production, Identification, Antioxidant Activity, and Stability. Mar Drugs. 2023, 21(3), 169.

--Line 81-85: Authors are advised to add references.

[1] Goginenia V, Hamannb MT. Marine natural product peptides with therapeutic potential: Chemistry, biosynthesis, and pharmacology. BBA - General Subjects 2018, 1862, 81-196.

[2] Islam Md S, Wang H, Admassu H, Sulieman AA, Wei FA. Health benefits of bioactive peptides produced from muscle proteins: Antioxidant, anti-cancer, and anti-diabetic activities. Process Biochemistry 2022, 116, 116–125

[3] Kong J, Hu XM, Cai WW, Wang YM, Chi CF, Wang B. Bioactive peptides from Skipjack tuna cardiac arterial bulbs (II): Protective function on UVB-irradiated HaCaT cells through antioxidant and anti-apoptotic mechanisms. Marine Drugs, 2023, 21(2), 105.

(5) Section Materials and Methods

The materials and methods involved are generally detailed and the processes are clear.

--However, in it is recommended that authors add references to experimental methods in this section.

-- The authors seem to have left out the materials and reagents. It is recommended that the author fill in the reagent contents, which are very important for the reader to repeat the experiment.

(6) Section Results

I find the results very complete, clear, and concise. However, How can the authors determine whether the synthesized peptide is the same compound as the same amino acid sequence in visfatin? Because amino acids also have different configurations.

(7) Section Discussion

--Line 315-320: This sentence is repeated with the paragraph in the Introduction (Line 81-87).

-- It is suggested that the results of this experiment should be compared with those of known peptides or other compounds with angiogenic activity.

6. PLOS authors have the option to publish the peer review history of their article (what does this mean?). If published, this will include your full peer review and any attached files.

Reviewer #1: No

Reviewer #2: No

---

## [Author Response · Author response to Decision Letter 0]

6 Jun 2023

Response to Editor-in-Chief

Is the manuscript presented in an intelligible fashion and written in standard English? 

Reviewer #1: No 

Reviewer #2: Yes

We thank you for this observation. We asked a native speaker of English to review our manuscript for syntax and grammatical errors and we have corrected typographical errors.

Response to reviewer #1 comments

1. There was no novelty in this study.

We thank you for this observation. Vascular endothelial growth factor (VEGF) and periostin have been well known to stimulate angiogenesis. However, some studies have demonstrated that the use of these full-length proteins has several inherent disadvantages, such as immunogenicity, lower stability, and loss of bioactivity (Lee et al., 2021). Therefore, some studies have reported the identification of VEGF- or periostin-derived angiogenic peptides containing about 10∼15 amino acids as an agonist to overcome the disadvantages of using full-length proteins (D’Andrea et al., 2005. Diana et al., 2008; Kim et al., 2017). However, to our knowledge, no studies on visfatin-derived angiogenic peptides have been carried out to date. In addition, the amino acid sequences of peptide-1 and -2 identified in the present study have not been reported in other studies. For this reason, we consider this to be a novel. Unfortunately, the reviewer felt that there was no novelty in this study, but if he can provide his logical reasoning for the same or previous studies, I will attempt to answer his concerns sincerely to his satisfaction. 

Lee K, Silva EA, Mooney DJ. Growth factor delivery-based tissue engineering: general approaches and a review of recent developments. J R Soc Interface. 2011; 8(55): 153–170. https://doi:10.1098/rsif.2010.0223.

D’Andrea LD, Iaccarino G, Fattorusso R, Sorriento D, Carannante C, Capasso D, et al. Targeting angiogenesis: Structural characterization and biological properties of a de novo engineered VEGF mimicking peptide. Proc Natl Acad Sci USA. 2005; 102 (40): 14215–14220. https:// doi: 10.1073/ pnas.0505047102.

Diana D, Ziaco B, Colombo G, Scarabelli G, Romanelli A, Pedone C, Fattorusso R, et al. Structural determinants of the unusual helix stability of a de novo engineered vascular endothelial growth factor (VEGF) mimicking peptide. Chemistry 2008, 14(14): 4164–4166. https:// doi: 10.1002/chem.200800180.

Kim BR, Kwon YW, Park GT, Choi EJ, Seo JK, Jang IH, et al. Identification of a novel angiogenic peptide from periostin. PLoS One. 2017 Nov 2;12(11):e0187464. https:// doi: 10.1371/journal.pone.0187464.

2. MD simulation should be required for docking validation

Thank you for pointing this out. We have considered this suggestion and performed the molecular dynamics simulations of visfatin-peptide complexes. We have included the required MD data in the materials and methods, results, and discussions sections of the revised manuscript.

3. Result and discussion section was not clear insight.

Thank you for pointing this out. We have revised the results and discussion section to clarify the insight from the study. 

4. Results were not consistent with previous studies.

We are grateful for this comment. However, like our present study, few studies have developed short length amino acid sequences (around 10) with angiogenic activity derived from the active site of the original full-sized visfatin using computer simulation technique. The previous studies have all isolated full-sized natural proteins or peptides from natural source or using recombinant DNA technology etc. Furthermore, the amino acid sequences of peptide-1 and peptide-2 in this study are very unique, and they have been first developed by this study, to the best of our knowledge. So, we could not compare our results with the results of previous studies. The reviewer has stated that the results were not consistent with previous studies. I would like to assure the reviewer that I will attempt to sincerely address the concern if details of such earlier studies are provided. 

. 

5. The biological unit of visfatin is dimer and why author did not consider dimer conformation for peptide design?

Thank you for pointing this out. We have performed molecular docking simulation to design peptides by considering the visfatin as a dimer in the GalaxyPepDock software and as a monomer in the HADDOCK software and calculated their corresponding scores. The highest-scoring two peptides obtained from the two methods above were considered for experimental analysis. 

Response to reviewer #2 comments:

Reviewer #2: In the manuscript (PONE-D-23-06800), authors researched the visfatin-derived angiogenic peptides by computer simulation approach. Overall, the topic is of scientific interest and would be of interest in the field of bioactive peptides. I think the manuscript can be accepted and published in PLOS ONE after a major revision. Suggestions are provided below:

(1) Title: Should be “Computer simulation approach to the identification of visfatin-derived angiogenic peptides” rather “Computer simulation approach to the identification of peptides with angiogenic activity based on the visfatin active site”.

Thank you for the suggestion. We have revised the title as suggested. 

(2) Keywords: Small peptide is not a standard term. I think that “oligopeptide” might be better than “small peptides”.

Thank you for pointing this out. We have made the change as suggested. 

(3) Section Abstract

--Line 34: Should be “Firstly” rather “First”. 

--Line 35: Should be “was” rather “was first”.

--Line 36: Should be “Secondly” rather “Second”.

We are grateful for this comment. We have made the change to the above.

(4) Section Introduction: The introduction provides an adequate background on the topic.

--Line 79-81: Authors are advised to add references.

--Line 81-85: Authors are advised to add references.

We thank you for this comment. We have changed this part to provide references as recommended as follows: 

However, high molecular weight of visfatin is a major limitation to its development as a therapeutic drug. In contrast, therapeutic peptides consisting of short-length amino acid sequence (3 to 20) have exhibited beneficial effects for the treatment of several health conditions (Lau et al., 2018; Wang et al., 2022). The peptides offer specific advantages such as good efficacy, good safety, low immunogenicity, high membrane permeability, and low cost compared to therapeutic proteins and antibodies (Fosgerau & Hoffmann, 2015; Muttenthaler 2021; Wang et al., 2022). For this reason, interest in the field of therapeutic peptides, including marine peptides, has increased in recent years. To date, more than 170 peptides are in active clinical development with many more in the preclinical stages (Research, 2016; Sridhar et al., 2021; Wang et al., 2022; Sheng et al., 2023). In particular, recent improvements in peptide screening and computational biology have increased the demand for peptide drug discovery (Wang et al., 2022). 

Lau JL, Dunn M. Therapeutic peptides: Historical perspectives, current development trends, and future directions. Bioorg Med Chem. 2018; 26(10): 2700-2707. https://doi: 10.1016/j.bmc.2017.06.052.

Wang L, Wang N, Zhang W, Cheng X, Yan Z, Shao G, et al. Therapeutic peptides: current applications and future directions. Signal Transduct Target Ther. 2022; 7(1): 48. https://doi: 10.1038/s41392-022-00904-4.

Fosgerau K, Hoffmann T. Peptide therapeutics: current status and future directions. Drug Discov Today. 2015; 20(1): 122-128. https://doi: 10.1016/j.drudis.2014.10.003.

Muttenthaler M, King GF, Adams DJ, Alewood PF. Trends in peptide drug discovery. Nat Rev Drug Disco. 2021; 20(4): 309–325. https://doi: 10.1038/s41573-020-00135-8.

Research TM. Global industry analysis, size, share, growth, trends and forecast. Pept. Mark. 2016–2024, 2016.

Sridhar K, Inbaraj BS, Chen BH. Recent developments on production, purification and biological activity of marine peptides. Food Res Int. 2021; 147: 110468. https://doi: 10.1016/j.foodres.2021.110468.

Sheng Y, Wang WY, Wu MF, Wang YM, Zhu WY, Chi CF, Wang B. Eighteen novel bioactive peptides from Monkfish (Lophius litulon) Swim Bladders: production, identification, antioxidant activity, and stability. Mar Drugs. 2023; 21(3): 169. https:// doi: 10.3390/md21030169. 

(5) Section Materials and Methods

The materials and methods involved are generally detailed and the processes are clear. -- The authors seem to have left out the materials and reagents. It is recommended that the author fill in the reagent contents, which are very important for the reader to repeat the experiment. 

We thank you for observation. We have changed several parts in the Materials and Method section to comply with your recommendation. 

(6) Section Results

I find the results very complete, clear, and concise. However, How can the authors determine whether the synthesized peptide is the same compound as the same amino acid sequence in visfatin? Because amino acids also have different configurations. 

One of the limitations of the present study is that it is uncertain whether the synthesized peptide had a configuration similar to visfatin as the same set of amino acids also had different configurations. Hence, we additionally evaluated the conformation and configuration of the synthetic peptides using circular dichroism (CD) spectroscopy. As shown in Fig. S1, peptide-1 was found to display two peaks (around 200 nm and 225 nm), and peptide-2 displayed a slight peak around 225 nm. These CD results did not show a typical spectrum of a-helix, b-sheet, and disordered form and therefore, we could not evaluate the configuration of the peptides. Nevertheless, the synthesized peptides may contain an optical isomer with the same configuration as that found in natural visfatin, and should have similar biological activities. In the future study, it would be important to consider the peptide configuration when designing peptides for therapeutic applications.

(7) Section Discussion

--Line 315-320: This sentence is repeated with the paragraph in the Introduction (Line 81-87).

We thank you for this observation. We have deleted the concerned sentence 

-- It is suggested that the results of this experiment should be compared with those of known peptides or other compounds with angiogenic activity.

We are grateful for this comment. We have introduced the following section in the discussion. 

It is well known that VEGF and periostin to stimulate angiogenesis. Therefore, some studies have reported the identification of VEGF- or periostin-derived angiogenic peptides containing about 10∼15 amino acids as agonists to overcome the disadvantages of using full-length proteins (D’Andrea et al., 2005. Diana et al., 2008; Kim et al., 2017). However, to our knowledge, no studies on visfatin-derived angiogenic peptides carried out to date. In addition, the amino acid sequences of peptide-1 and -2 identified in the present study have not been reported earlier and hence this study may be considered a novel study.

D’Andrea LD, Iaccarino G, Fattorusso R, Sorriento D, Carannante C, Capasso D, et al. Targeting angiogenesis: Structural characterization and biological properties of a de novo engineered VEGF mimicking peptide. Proc Natl Acad Sci USA. 2005; 102 (40): 14215–14220. https:// doi: 10.1073/ pnas.0505047102.

Diana D, Ziaco B, Colombo G, Scarabelli G, Romanelli A, Pedone C, Fattorusso R, et al. Structural determinants of the unusual helix stability of a de novo engineered vascular endothelial growth factor (VEGF) mimicking peptide. Chemistry 2008, 14(14): 4164–4166. https:// doi: 10.1002/chem.200800180.

Kim BR, Kwon YW, Park GT, Choi EJ, Seo JK, Jang IH, et al. Identification of a novel angiogenic peptide from periostin. PLoS One. 2017 Nov 2;12(11):e0187464. https:// doi: 10.1371/journal.pone.0187464.

---

## [Editor Report · Decision Letter 1]

9 Jun 2023

Computer simulation approach to the identification of visfatin-derived angiogenic peptides

PONE-D-23-06800R1

Dear Dr. Joo,

We’re pleased to inform you that your manuscript has been judged scientifically suitable for publication and will be formally accepted for publication once it meets all outstanding technical requirements.

Kind regards,

Belgin Sever, Ph.D.

Academic Editor

PLOS ONE
---

## [Editor Report · Acceptance letter]

14 Jun 2023

PONE-D-23-06800R1 

Computer simulation approach to the identification of visfatin-derived angiogenic peptides 

Dear Dr. Joo:

I'm pleased to inform you that your manuscript has been deemed suitable for publication in PLOS ONE. Congratulations! Your manuscript is now with our production department. 

Kind regards, 

on behalf of

Assoc. Prof. Dr. Belgin Sever 

Academic Editor

PLOS ONE